# Comparison of Two Chosen 3D Printing Resins Designed for Orthodontic Use: An In Vitro Study

**DOI:** 10.3390/ma16062237

**Published:** 2023-03-10

**Authors:** Anna Paradowska-Stolarz, Joanna Wezgowiec, Marcin Mikulewicz

**Affiliations:** 1Division of Dentofacial Anomalies, Department of Orthodontics and Dentofacial Orthopedics, Wrocław Medical University, Krakowska 26, 50-425 Wrocław, Poland; 2Department of Experimental Dentistry, Wrocław Medical University, Krakowska 26, 50-425 Wrocław, Poland

**Keywords:** 3D printing in dentistry, dental materials, tensile, compression, resins

## Abstract

(1) Background: In recent years, 3D printing has become a highly popular tool for manufacturing in various fields such as aviation, automobiles, plastics, and even medicine, including dentistry. Three-dimensional printing allows dentists to create high-precision models of teeth and jaw structure, and enables them to develop customized tools for patients’ treatment. The range of resins used in dentistry is quite large, and this branch is developing rapidly; hence, studies comparing different resins are required. The present study aimed to compare the mechanical properties of two chosen resins used in dentistry. (2) Materials and methods: Ten specimens each of two types of 3D-printable resins (BioMed Amber and IBT, developed by Formlabs) were prepared. The samples were printed on a Formlabs Form 2 3D printer according to ISO standards. Samples for the compression test were rectangular in shape (10 ± 0.2 mm × 10 ± 0.2 mm × 4 ± 0.2 mm), while the samples used for the tensile test were dumbbell shaped (75 mm long, with 10 mm end width and 2 mm thickness). Tensile and compression tests of both materials were performed in accordance with the appropriate ISO standards. (3) Results: The BioMed Amber resin was more resistant to compression and tensile forces, thus implying that the resin could withstand higher stress during stretching, pulling, or pushing. The IBT resin was less resistant to such loads, and failure of this material occurred at lower forces than those for Biomed Amber. An ANOVA test confirmed that the observed differences were statistically significant (*p* < 0.001). (4) Conclusions: Based on the properties of both materials, the IBT resin could be better used as a tray for placing orthodontic brackets through an indirect bonding technique, while the BioMed Amber resin would be more useful as a surgical guide for placing dental implants and mini-implants. Further potential fields of application of the resins should be investigated.

## 1. Introduction

Three-dimensional (3D) printing has become an increasingly popular manufacturing tool in the medical world. It has many applications, and is particularly used in the printing of surgical guides, custom parts, and anatomical models. These 3D-printed pieces are widely used in many types of surgeries, especially in bone surgery. These pieces fit precisely in their predetermined place, thereby reducing the time required in surgical procedures. Printable materials have similar properties to traditional ones, but they ensure more accuracy, detailed shape, and fast performance. Unfortunately, scanning and additional prints increase the cost of printing procedures [1]. The properties of the materials also depend on the position of the layers and the weave structure of the fabricated piece [2,3].

Presently, 3D printing is commonly applied in dentistry to fabricate a variety of dental parts. The materials used in medicine, including dentistry, must be biocompatible, which means that they must be neutral, nonirritating, and noncytotoxic to the surrounding tissues [4]. Moreover, treatment planning requires the use of cone-beam computerized tomography (CBCT) for visualizing the surrounding tissues and then planning the surgical procedures. Individual splints can be used as surgical guides for implant placement and orthodontic mini-implant placement [1,5]. These surgical guides were first introduced in general surgical procedures, including dental implants, where there is a risk related to a small amount of space between the roots, which could lead to root damage during implant placement. Later, the guides were incorporated into other procedures, such as orthodontic mini-implant placement [6]. The guide is planned based on the CBCT image [7]. To correctly position the implant or orthodontic mini-implant, special software and the possibility of creating “standard tessellation language” (STL) casts are required [8]. The precision of implant placement depends not only on the CBCT scan and surgeon experience, but also on the number of teeth present in the oral cavity, particularly in the surrounding area of the implant site. Due to anatomical issues, molars and premolars provide greater precision than incisors and canines [9].

In orthodontics, surgical guides are most commonly used for the placement of mini-implants, which is crucial, particularly when orthognathic surgery is planned. The surgical guides can be used before the placement of the hybrid hyrax expander and before MARPE (maxillary palatal expansion) planning [10]. The novel implant placement technique assumes the use of surgical guides for greater accuracy and a reduction in the risk of failure in implant placement. This refers to both tooth implants and orthodontic mini-implants. The use of a surgical guide during implant placement generates higher temperatures (>42 °C) than without it; consequently, this could cause some unexpected changes in bone structure. This could influence bone healing. Hence, whether the use of a surgical guide is mandatory or not should be considered. If the space where the implant or mini-implant is inserted is small, the use of a surgical guide is very helpful for the precise placement of the device, to avoid any damage to the surrounding teeth, particularly to their roots [11].

In summary, 3D techniques are being used more frequently in modern dentistry, including orthodontics. Apart from the most popular application for the clear preparation of aligners, they can be used for additional purposes such as splints for surgical guidance, occlusal splints, individual orthodontic appliances, and/or orthodontic bracket trays [12]. However, the proper selection of the 3D-printable material is necessary for the successful utilization of this technique. Hence, the properties of the materials available must be carefully investigated before selection for a specific application.

## 2. Aim of the Study

The first aim of the present study was to compare the mechanical properties (compression and tensile moduli) of two 3D-printable resins (BioMed Amber and IBT) used in dentistry, including orthodontics and surgery. The authors investigated these mechanical features to assess the durability of the chosen materials. The second aim of this study was to propose the material that would be better for producing surgical guides. Thus far, few studies have compared the properties of different 3D-printed materials. To the best of the authors’ knowledge, a comparison of these two resins has not yet been conducted. This paper is part of a planned series of papers on the different features of 3D printing materials.

## 3. Materials and Methods

### 3.1. Materials

Two 3D-printable resins, namely BioMed Amber (Formlabs, Ohio, Milbury, OH, USA) and IBT (Formlabs, Ohio, Milbury, OH, USA), were tested to evaluate their properties. Both are biocompatible materials, but with different properties. The BioMed Amber resin is a rigid and strong material, compatible with sterilization and disinfection agents, while the IBT resin is elastic and flexible. Table 1 presents the properties of the materials, according to the producer, and their potential use.

### 3.2. Preparation of Specimens

Material blocks were printed using a Form 2 3D printer (Formlabs, Sommerville, MA, USA), with violet light at 405 nm and 250 mW. The samples were prepared according to the appropriate ISO standards and instructions of the manufacturer. The printer is self-adjustable, and the settings were adjusted during the placement of the resin cartridge with a built-in chip. The print layer was 100 microns. The printing temperature was maintained at around 35 °C.

Two types of blocks were used to measure the selected properties: a rectangular block, according to the ISO 604:2003 standard (for the compression test), and a dumbbell-shaped block (type 1BA), according to the ISO 527-1:2019(E) standard (for the tensile test) [13,14]. According to ISO standards, the minimum number of probes in this kind of research should be five. Therefore, 10 specimens of each type and each material were printed on the Form 2 printer and calibrated for medical use. The resins were in their original packaging and the cartridge was opened just before the start of the investigation [15]. The printer was set automatically by a built-in chip in the cartridge. After printing, the specimens were rinsed twice in isopropanol (Stanlab, Lublin, Poland) for 10 min each. Following rinsing, the specimens were dried at room temperature for 30 min. Subsequently, post-curing with Form Cure (Formlabs) was performed for 30 min at 60 °C for the BioMed Amber resin and for 60 min at 60 °C for the IBT resin, and the supports were then removed. The samples, thus prepared, were incubated at room temperature and 50% relative humidity (RH) for 24 h (for the tensile test) or 4 days (for the compression test), and then subjected to the tests. The properties of these two materials were assessed using 10 blocks for each test. The tests were conducted at room temperature (ca. 24.3 °C) and humidity of 36.6–37.8% for two successive days, so that the conditions would be comparable.

### 3.3. Compression Test

To reduce the risk of measurement failure, the width and height of the samples were measured at five points by using a Magnusson digital caliper (150 mm) (Limit, Wroclaw, Poland). The mean values were then calculated. The axial compression test was performed according to the PN-EN ISO 604:2003 standard by using a Universal Testing Machine (Z10-X700, AML Instruments, Lincoln, UK) at a speed of 1 mm/min. The performed measurements allowed us to calculate the compression modulus. Figure 1 presents all the information about the test. The sample tested in the Universal Testing Machine is shown in Figure 2A.

### 3.4. Tensile Test

To reduce the risk of measurement failure, the width and height of the samples were measured at five points by using a Magnusson digital caliper (150 mm) (limit). The mean values were then calculated. The tensile test was performed using a Universal Testing Machine (Z10-X700, AML Instruments, Lincoln, UK) at a constant crosshead speed of 5 mm/min. The specimens that broke outside the test length were identified. The tensile modulus for both materials was measured. All the information is presented in Figure 3. The sample tested in the Universal Testing Machine is shown in Figure 2B.

### 3.5. Statistical Analysis

The Shapiro–Wilk normality test was performed to determine the type of statistical test to use for the obtained data. Figure 4 and Figure 5 present histograms of the statistical data.

The Mann–Whitney *U* test was used to compare Young’s moduli of the resins. The results are presented in a table and as box plots showing the summary of a set of data in terms of five values (minimum, first quartile, median, third quartile, and maximum). The number of samples in each group was 10. All statistical data were analyzed using the program Statistica v. 13 (TIBCO Software Inc., Palo Alto, CA, USA).

## 4. Results

The results of the study are presented in Table 2 and Figure 6.

Due to the high sensitivity of the Shapiro–Wilk normality test, particularly for small sample-size investigations, the authors decided to use the Mann–Whitney *U* test for result presentation.

Table 2 presents the basic statistical data on the mechanical properties (compression and tensile moduli) of the two tested resins. The results showed that the BioMed Amber resin was significantly more resistant to both compression and tensile moduli than the IBT resin (*p* < 0.001 for both presented results).

Figure 6 shows the graphical comparison of compression and tensile moduli, and the results of the Mann–Whitney *U* tests for both resins.

The obtained data revealed that the IBT resin was significantly more susceptible to breakage in both the compression and tensile tests. BioMed Amber was more resistant to these forces than the IBT resin. Thus, BioMed Amber is more rigid and durable than IBT. BioMed Amber showed high resistance in both tests, while IBT resin cracked easily and showed minimal resistance in the mechanical tests.

## 5. Discussion

Both of the tested 3D-printable materials, IBT and Biomed Amber, are designed for use in medical applications, including various fields of dentistry. According to the recommendations of the producers, one of the uses of these resins is in orthodontics and orthodontic devices. It is, however, debatable whether both these resins could be used for the same purposes. In addition to these applications, these materials could be used in other branches of dentistry; for example, in surgery for the production of surgical guides. The results of the present study showed that IBT is less stable and much more susceptible to damage or deformation than Biomed Amber. Therefore, it remains questionable whether IBT should be used for precise surgical guidance. The risk of complications due to material instability would be quite high when using IBT [16,17].

As reported earlier [18], 3D printing is an acceptable method, but it does not ensure 100% accuracy. This especially refers to the type of treatment planning when more than one device, for example, a splint, is required. Therefore, this phenomenon could influence the effectiveness of orthodontic treatment with clear aligners. Due to the fact that the position of teeth changes and each splint carries an error in it, the appliance, in this case, may not fit the teeth accurately when progressing to further steps of treatment. Furthermore, the planned thickness of the appliance is lower than that of the accurately printed material, which results in an open bite at the end of orthodontic treatment and the need for final improvements [19]. Some orthodontic appliances are prepared only by a group of specialists for cleft patients. In this case, the accuracy of the scan and print is crucial when obturators are prepared. This kind of appliance is prepared to help feed the patient, as well as to mold the clefted maxilla and prepare it for surgery. Three-dimensional printing technology has made this process easier and more accurate [20,21].

External factors influencing the appliance can also change the properties of the materials. One of these external factors is food and beverages. Warnecki et al. [22] confirmed that acidic drinks, such as coca cola and orange juice, can damage the structure of the clear aligners’ splints. Damage to the structure would impair tooth movement. Although this type of appliance should be removed while eating, removing it each time the patient needs something to drink could be problematic. There are some new perspectives in the preparation of 3D materials. The use of nanostructured materials could usher in a new era of 3D-printing accuracy, which is a promising future perspective [23].

Most 3D-printable materials, and their modifications, are used in prosthetics; namely, to fabricate removable dentures and occlusal splints. Due to the rapid development of these types of materials and their applications, there has been an increase in use of dental polymers in other branches of dentistry as well [24]. In addition to prosthodontics and orthodontics, these materials are used in conservative and cosmetic dentistry to restore hard tissue loss [2]. However, to ensure successful results for specific applications, proper material selection is required based on the analysis of their properties.

Based on its flexibility, the IBT resin could be successfully used as a tray for indirect bracket placement, although our research showed that it should not be used to prepare more stable and precise products, such as surgical guides [16,25]. For an indirect bonding tray, this kind of resin shows sufficient accuracy and low linear and angular deviations, which provides good reflection in the final bracket and/or attachment bonding [26].

Due to its rigidity and stiffness, the BioMed Amber resin could be used to produce more precise elements, including surgical guides, for implant and orthodontic mini-implant placement. As shown in our study, this resin is durable and has low distortion values. Investigation of the other properties of this resin would be a valuable approach to indicate its other potential uses. The comparison of the BioMed Amber resin with other type of 3D-printed resin, in the authors’ previous study, revealed that the former is more resistant to compression than tension, as compared to another rigid resin, Dental Clear LT [27]. As the Biomed Amber resin could be used for surgical mini-implant placement, it could also be used in the preparation of hybrid hyrax expander placement. This kind of appliance allows more skeletal changes than the regular one, which was previously introduced to orthodontics. A hybrid hyrax needs mini-implants to anchor it to the palatal bone, and the placement of these mini-implants should be very precise; otherwise, the appliance would not fit [28,29]. For orthodontic uses, a wide range of appliances, such as highly esthetic stabilizers for Class II malocclusion, could also be prepared with this technique [30].

## 6. Conclusions

IBT is more adequate for use in an indirect bonding tray, while BioMed Amber could be used more as a stiff material; for example, as a surgical guide in implant and mini-implant placement.

## 7. Limitations

The current study has some limitations. The number of samples was small (*n* = 10), although it still meets ISO standards. As the authors consider this as the biggest limitation of the present study, the number of samples should be increased in future tests. Additional tests, such as flexural characterization of the materials, could also be considered. We did not incorporate this test in our present research because it is mainly used when restorative resins are tested; however, this test could be considered to widen the scope of future research on this feature [31]. Furthermore, the general use of the studied materials is limited because they are intended only for short-term use. However, according to the producer, the BioMed Amber resin could be sterilized, using the standard method, and reused. This is a very important aspect, especially when planning surgical procedures, because it reduces the possibility of tissue contamination. On the other hand, it remains doubtful whether IBT resin could be used after sterilization because of its potential fragility. The IBT material showed low resistance to tensile and compression tests, even without sterilization or disinfection. Hence, further studies on this topic are required. For such materials, other methods of sterilization, such as UV light or ozone, could be considered. Ozone is a promising method of disinfection because it minimally affects the properties of other materials [15]. However, one must be aware that the success of decontamination also depends on the surface of the disinfected or sterilized structure [32]. In this case, the IBT resin could be problematic because of the high possible roughness of the nonstable material. The use of coatings, such as chitosan, to prevent contamination could also be considered [33]. In this case, the use of natural polymers is highly desired because of their biocompatibility and abundance in the environment. It is also known that the properties of these materials deteriorate over time, not only because of external conditions, but also due to material wear. This phenomenon also occurs in 3D-printable dental materials [22,34]. Although this was not the subject of the present investigation, future investigations should consider these aspects as an interesting path in this developing branch of dentistry.

## Figures and Tables

**Figure 1 materials-16-02237-f001:**
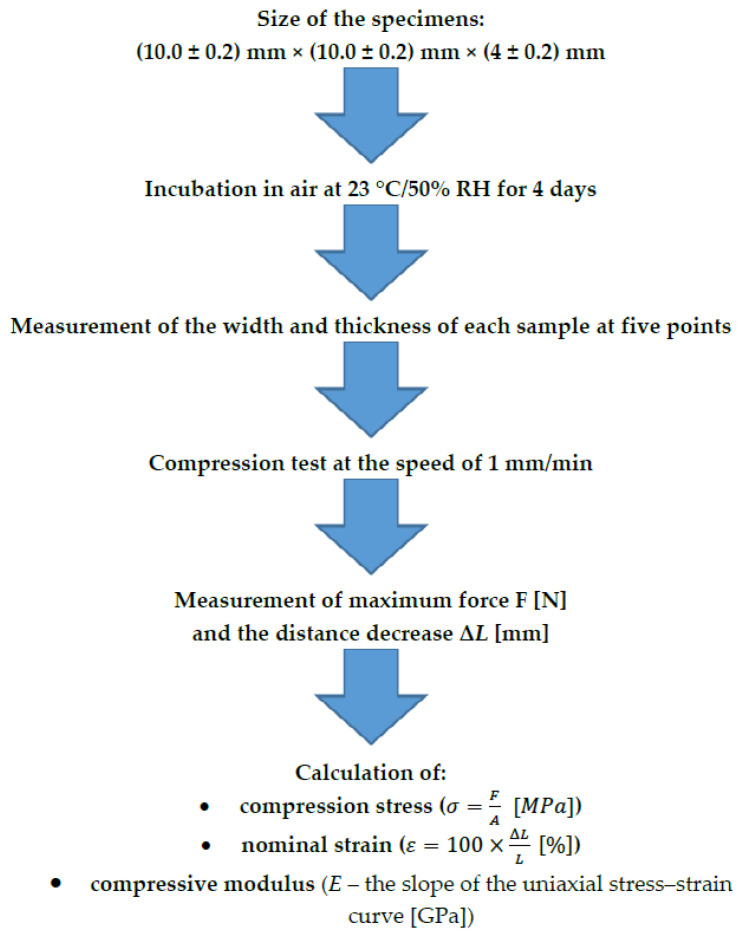
Axial compression test according to the PN-EN ISO 604:2003 standard (*F*, force (N); *A*, initial cross-sectional area measurement (mm^2^); *L*, measurement of the distance between compression plates (mm); Δ*L*, decrease in distance between the plates (mm)).

**Figure 2 materials-16-02237-f002:**
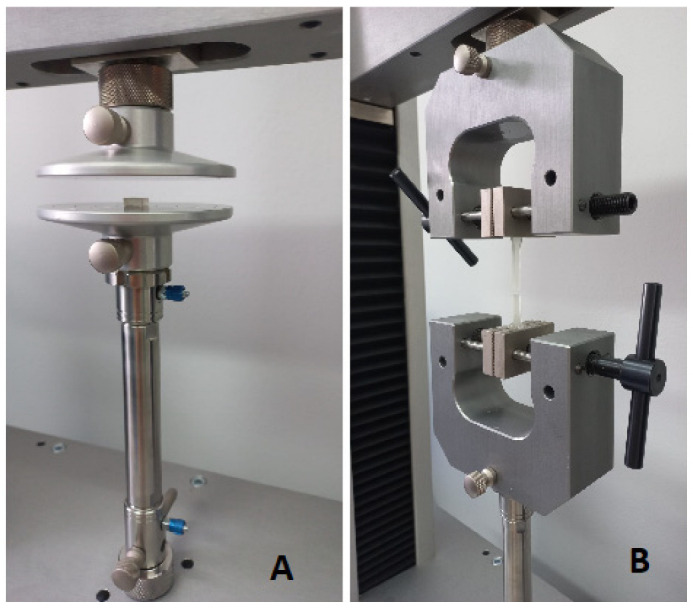
Illustration of the samples tested in the Universal Testing Machine during the (**A**) compression test and (**B**) tensile test.

**Figure 3 materials-16-02237-f003:**
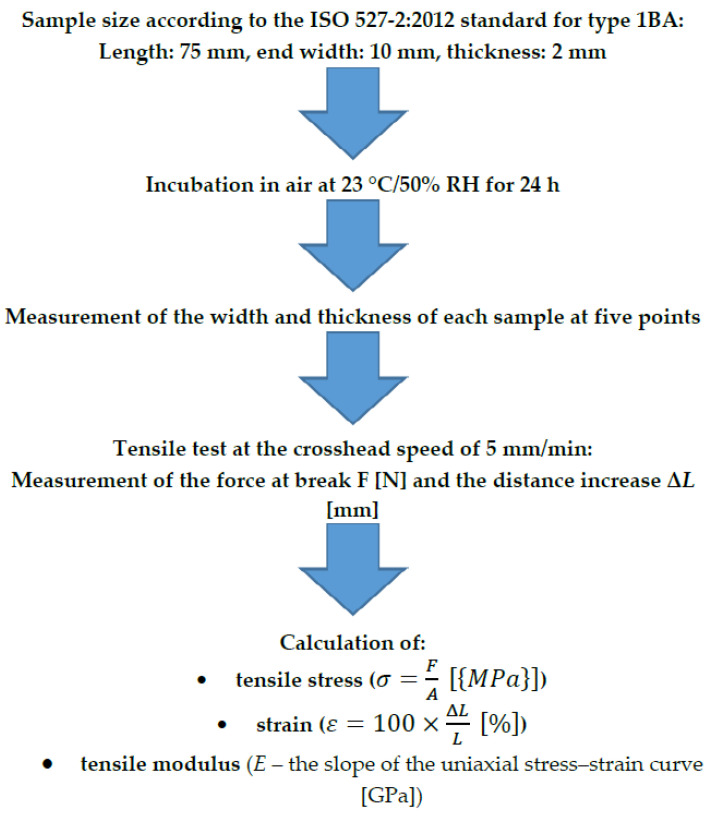
Tensile test according to the PN-EN ISO 527-1: 2019 (*E*) standard (*F*, force (N), *A*, initial cross-sectional area measurement (mm^2^); *L*, measurement of the distance between the grips (mm); Δ*L*, increase in distance between the grips (mm)).

**Figure 4 materials-16-02237-f004:**
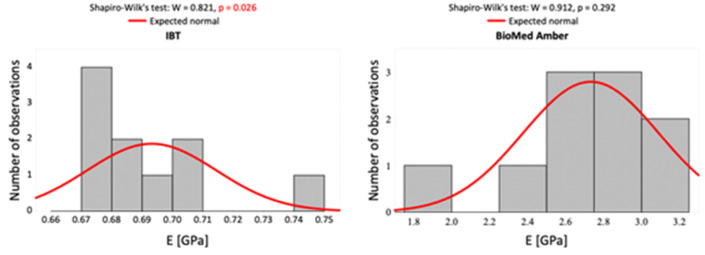
Histograms of the measurement results of Young’s modulus in the compression test against a background of normal data distribution for the two dental resins and the normality test results.

**Figure 5 materials-16-02237-f005:**
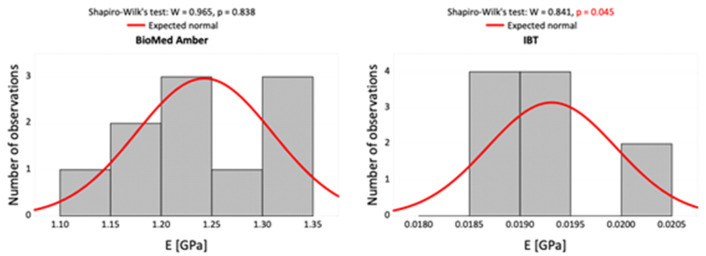
Histograms of the measurement results of Young’s modulus in the tensile test against a background of normal data distribution for the two dental resins and the normality test results.

**Figure 6 materials-16-02237-f006:**
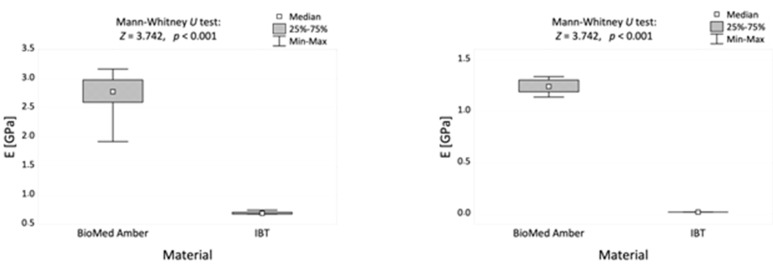
Compressive modulus (**left**) and tensile modulus (**right**) of the two dental resins and the results of the Mann–Whitney *U* test.

**Table 1 materials-16-02237-t001:** Properties of BioMed Amber and IBT resins (according to the producer).

	BioMed Amber Resin	IBT Resin
Biocompatibility	- Biocompatible - For short-term skin and mucosal contact	- Biocompatible—noncytotoxic, nonallergic, nonirritating - For short-term skin and mucosal contact
Physical properties	- Rigid - Strong - Transparent, beige	- Elastic, flexible - Translucent, transparent - Optimized tear strength
Compliance with all standards	- Compatible with common disinfection and sterilization agents - ISO 13485 - EN ISO 10993-5:2009 - ISO 10993-10:2010/(R)2014 - ISO 10993-10:2010/(R)2014	- EN ISO 10993-5:2009 - ISO 10993-10:2010/(R)2014 - EN ISO 13485:2016 - EN ISO 14971:2012
Use	- Strong, right parts - Surgical guides - Models to access accurate implant size - Sample collection kits - Guides for cutting and drilling - Functioning threads - Parts of medical devices	- Indirect bonding trays - Tooth templates - Orthodontic/surgical guides

**Table 2 materials-16-02237-t002:** Basic statistical data of the mechanical properties of the two dental resins.

Young’s Modulus (GPa)	Resin		*p*-Value
	BioMed Amber	IBT	
	*n* = 10	*n* = 10	
Compression			<0.001
Me (Q1; Q3)	2.78 (2.60; 2.98)	0.69 (0.68; 0.71)	
Min–Max	1.92–3.17	0.67–0.74	
Tensile			<0.001
Me (Q1; Q3)	1.24 (1.19; 1.30)	0.019 (0.019; 0.019)	
Min–Max	1.14–1.34	0.019–0.020	

Me—median, Q1—lower quartile, Q3—upper quartile, *p*—the level of significance for the Mann-Whitney *U* test.

## Data Availability

All detailed data could be found at authors A.P.S. and J.W.

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
