# Peer review of "Comparison of Two Chosen 3D Printing Resins Designed for Orthodontic Use: An In Vitro Study"

_materials, 2023, doi:10.3390/ma16062237_

Round 1
Reviewer 1 Report (Previous Reviewer 2)
The manuscript discusses a current area of study that has implications for the usage of BioMed Amber and IBT, two 3D printable resins, in orthodontics (Formlabs, USA). The manuscript also discusses the two selected dental resins' characteristics. Both resins have unique characteristics despite being biocompatible. The research develops recommendations for the usage of each type of resin based on the outcomes of the evaluation of their properties, which is what makes the study very good: IBT resin's qualities indicate that it might be used more effectively as a tray for the indirect bonding technique used to bond (place) orthodontic brackets, whereas BioMed Amber would be more useful as a surgical guide for the placement of dental implants.
Author Response
Dear Reviewer 1,
thank you for the positive feedback and summing up the article. Thank you for an effort in reviewing our paper. Your opinion is a very important to us. Thank you once again for the time you spent on reviewing this paper. Best regards.
Reviewer 2 Report (Previous Reviewer 3)
The title of the work does not have the correct syntax. The abstract structure does not include headings of specific chapters such as background material, etc. There are several limitations that can be noted in this scholarly article, including:
-
Small sample size: Authors have properly discussed that their study used only 10 specimens of each resin, which may not be sufficient to provide a comprehensive understanding of the materials' properties. A larger sample size would increase the statistical power and reduce the likelihood of obtaining misleading results.
-
Limited generalizability: The study was conducted using only two specific types of resins, and the results may not be applicable to other resins used in dentistry - orthodontic applications. Additionally, the study only tested mechanical properties and did not consider other factors, such as biocompatibility or durability.
-
Lack of information on the testing conditions: The article mentions that the specimens were printed and tested according to ISO standards, but it does not provide details on the specific testing conditions or the equipment used. This lack of information makes it difficult for other researchers to replicate the study or compare the results to other studies.
-
Limited scope: The study focused on only two specific applications of the resins (orthodontic brackets and surgical guides for dental implants), and the authors themselves point out that further research is needed to identify other potential applications. Therefore, the conclusions of the study should be interpreted with caution and should not be generalized beyond the applications tested.
On page four, figure one, the blue arrows should appear slightly smaller and without the bold text.
Authors have used normality test (Shapiro-Wilk). Main limitation of this normality test, specifically the Shapiro-Wilk test, is that it is sensitive to sample size. In large samples, even small deviations from normality can lead to a rejection of the null hypothesis of normality, while in small samples, the test may fail to detect non-normality even when it is present. As a result, the test may provide misleading results and lead to incorrect inferences when used inappropriately. Additionally, it is important to elaborate in methods that normality is often not strictly required for many statistical methods to be valid, especially in larger samples. Therefore, the use of normality tests should be supplemented with other diagnostic methods and careful consideration of the underlying assumptions and the context of the analysis.
The discussion revolves around the use of different types of 3D-printable dental resins, particularly IBT resin and BioMed Amber resin, in prosthodontics, orthodontics, conservative and cosmetic dentistry. The authors researched the properties of these materials to determine their suitability for specific applications such as the fabrication of surgical guides and indirect bonding trays for orthodontic treatment. The research found that IBT resin is more flexible and suitable for less stable and precise products, while BioMed Amber resin is more rigid and suitable for more precise elements. The authors shall discuss other applications of 3D printing with resins in orthodontics, e.g., palatal plates in patients with cranial syndromes such as the Pierre Robin sequence - Using optical scanning and 3D printing to fabricate custom appliances for patients with craniofacial disorders DOI 10.1053/j.sodo.2022.10.005 or Pierre Robin sequence and 3D-printed personalized composite appliances in an interdisciplinary approach DOI
10.3390/polym14183858
Authors note that the current study had limitations, such as a small sample size, and suggest that further research should be conducted on other aspects of these materials, such as their properties over time and their suitability for sterilization and disinfection.
Since the Discussion chapter is quite long, I would suggest creating a separate Conclusions chapter with a single paragraph or bulleted list summarizing the conclusions of the paper.
After corrections I find this paper useful and suitable for publication
kind regards
Author Response
Dear Reviewer, thank you for the detailed report. See the attached file with responses to your comments. Best regards

Reviewer 3 Report (New Reviewer)
Dear Authors
The methodology presents some lacks and weaknesses.
The English language is often distracting and poor including orthographic errors.
TITLE
It does not describe the intent of the work. It could be better like this: “Comparison of two 3D printing resins designed for orthodontic use: an in vitro study
ABSTRACT
The aim should be better described and above all, it should emerge from the shortcomings found on the subject in the literature
The MATERIALS AND METHODS section has some lacks:
Please clarify the tests as they are performed and under what environmental conditions
DISCUSSION
This section should be better developed. It seems just a repetition of the results, only with a description of the data.
REFERENCES
are limited to those necessary to support the study but there are some typographical mistakes such as missing volumes or pages and punctuation marks. Please consider adding these references to yours: 10.3390/dj10070132
- and PMID: 24745595
Author Response
Dear Reviewer 3,
thank you for an effort. We attached the file with all the responses to your suggestions. We hope we fulfilled your requirements.
Best regards - Authors

Reviewer 4 Report (New Reviewer)
The data are not enough for a standard scientific paper as the authors have considered only a single outcome variable i.e. elastic modulus.
The novelty is completely lacking. The comparison of elastic modulus of two 3D printing resin cannot justify the novelty and originality of this paper.
Why did the authors only evaluate elastic modulus?
Why did the authors test the specimens in tensile and compression modes? Are these both modes relevant to clinical scenarios?
For the polymers, ISO 4049 has advocated the testing of specimens in the flexural mode, why did authors not consider flexural characterization?
The manufactures of both resins have stated bold claims regarding the strength, flexibility, translucency, biocompatibility etc. If authors wanted to validate the claims of the manufactures then they should have considered all characteristics in their research work.
In the current form, this paper looks like a short technical report. It is suggested that the authors should consider the detailed evaluation of all relevant in vitro properties of both resins so as to elucidate their behavior in the scientific manner.
Author Response
Dear Reviewer,
the response is incorporated to the attached file. Please, see the attachment.
Best regards and thank you once again for the effort and time you spent on the review of our paper. Hopefully, incorporated changes would satisfy you - Authors.

Round 2
Reviewer 2 Report (Previous Reviewer 3)
Authors have improved the manuscript sufficiently.
Author Response
Thank you for the support, I am attaching the certificate of English corrections. Thank you - Authors

Reviewer 4 Report (New Reviewer)
The authors have provided responses to my comments, however, I do not find them satisfactory.
In the current form, data are not adequate to meet the standard of publication in the 'Materials'.
It is suggested that the authors should add more variables and present the paper with scientific mechanisms.
The variables for instance strength, translucency and biocompatibility will be of great value.
Author Response
Dear Reviewer,
we are sad to hear that the corrections did not meet the cryteria of the Reviewer.
Additional variables would change the whole manuscript and we are not able to prepare the presented tests at this stage of the research (I discussed that with the laboratory). We would like to widen the project in the future though.
Additionally, we add the certifficate that this version was prooread by professional translator.
With great respect - Authors
This manuscript is a resubmission of an earlier submission. The following is a list of the peer review reports and author responses from that submission.
Round 1
Reviewer 1 Report
The manuscript addresses a modern research topic with an impact on the use of two 3D printable resins in orthodontics: BioMed Amber and IBT (Formlabs, USA). Also, the manuscript present the properties of the two chosen resins used in dentistry. Although both resins are biocompatible, they have different properties. The originality of the study is represented by the fact that the research establishes recommendations for the use of each type of resin, based on the results of the evaluation of their properties: “The properties of both of the materials show that IBT resin could be used better as a tray for orthodontic bracket placement in indirect bonding technique while as BioMed Amber would be more useful as a surgical guide for dental implants and miniimplants placement. Further potential uses of the resins should be investigated.”
The manuscript addresses a modern research topic with an impact on the use of two 3D printable resins in orthodontics: BioMed Amber and IBT (Formlabs, USA). Also, the manuscript present the properties of the two chosen resins used in dentistry. Although both resins are biocompatible, they have different properties. The originality of the study is represented by the fact that the research establishes recommendations for the use of each type of resin, based on the results of the evaluation of their properties: “The properties of both of the materials show that IBT resin could be used better as a tray for orthodontic bracket placement in indirect bonding technique while as BioMed Amber would be more useful as a surgical guide for dental implants and miniimplants placement. Further potential uses of the resins should be investigated.”

Reviewer 2 Report
Thanks for asking me to review this manuscript.
But I believe this paper requires extensive English language editing and improvements regarding clarity and readability.
My comments are as follows:
Title
1- This title, "Comparison of 2 3-D printing resigns used in orthodontics," requires clarifying the outcomes being measured and the two materials being tested. In addition, the authors should make clear that they are testing these materials in vitro. Also, you are studying resins and NOT "RESIGNS". Please avoid all mistakes in your writing up.
Abstract
2- Please expand the abstract to reach 200 to 250 words. You need to insert more information. The Results section is almost empty. You need to insert the mean (or median) values from your statistical analysis. You need to make your Abstract full of information.
Introduction
3- I could not understand the Introduction section very well. There are several problems regarding grammar, spelling, punctuation, and style. The writing up of this section and the other sections required extensive editing and re-phrasing. Please consider using special companies specialized in research writing up editing.
4- Some reference should be cited here, such as:
- Jaber ST, Hajeer MY, Khattab TZ, Mahaini L. Evaluation of the fused deposition modeling and the digital light processing techniques in terms of dimensional accuracy of printing dental models used for the fabrication of clear aligners. Clin Exp Dent Res. 2021 Aug;7(4):591-600. doi:10.1002/cre2.366. Epub 2020 Nov 30. PMID: 33258297; PMCID: PMC8404487.
- Hajeer MY, Millett DT, Ayoub AF, Siebert JP. Applications of 3D imaging in orthodontics: part II. J Orthod. 2004 Jun;31(2):154-62. doi: 10.1179/146531204225020472. PMID: 15210932.
5- The justification for the onset of this research has not been given. The last paragraph does not prepare the readers for a reason beyond conducting this research.
Materials and Methods
6- Not all mentioned brands, products, and materials are accompanied by the correct information about the company name, city, and country.
7- Noramitly tests should not be commented on elaborately. The authors are not required to present the results of the normality tests. They should speak briefly about the related results and then move on to the next steps.
8- In Figures 1 and 2, the authors are asked to explain all the abbreviation mentioned in these figures.
9- Figures 5, 6, and 7 are not required. Suppose all the pieces of information (i.e., the whole results) are available in the tables; these figures are no longer needed.
Good luck with the revised version of your paper.
Reviewer 3 Report
Regarding the title
please Rewrite the title according to the instructions to authors and common criteria for scientific articles. The title is misleading, and the term orthodontics shouldn't be forced upon this paper.
In the abstract there are none as existing words used for example “ resigns”
also in the keywords these non-existing words are used ”resigns” authors probably mean the word resins Please revise the abstract and also keywords in the proper way. Composite keywords such as “3-D printing in dentistry” using dash is not correct.
In the paper, there are many syntax error English needs to be revised significantly. Unfortunately, there are also many terminological errors. “hYbrid-hyrax placement”. Others should first use the full expression followed with abbreviation for example on the page #2. Others have turned off the line numbering in the template as is prescribed that is difficult to conduct this review. The lines number shall be visible on the right side of the page.
It's not quite clear how the authors mean the utilization of those two resins in orthodontics as they mentioned surgical guides for orthognathic surgeries or surgical placement of the TAD temporary anchorage device. This is known and the authors also repeat both resins are intended only for a short-term skin and mucosal contact, then the true utilization in orthodontics as long-term splint or aligner is not possible. Utilizations of both resins in orthodontics is quite limited to temporary bonding trays or other temporary and short-term exposures to oral cavity. The most frequent use for those two is surgically-related application *for surgical guides in orthognathic surgeries or implant placement which shall not be considered orthodontic domain purpose only.
Regarding the discussion
It is not clear what led authors to the conclusion in the first sentence of discussion that “Both printable materials, IBT and Amber could be used in dentistry, especially in orthodontics.” Despite both materials can be used in orthodontics, they are used is quite limited. Orthodontic applications are rather long term affect applied in oral cavity and for such there are more appropriate by a compatible results applicable for unlimited time in the oral cavity.
This paper is of poor scientific quality and the main concept of the paper is confusing. If the paper compares two materials of biocompatible resins for general use in dentistry it shall be more suitable. The number of samples analyzed in the paper was too small and the interpretation including the terminology is not appropriate for scientific. This paper needs significant rewriting and fundamental revision. It shall definitely reference some recent papers researching utilization of Long term by compatible resins in orthodontics.
In general, the paper lacks the quality and level of Q1 scientific journal.